# The neural representation of absolute direction during mental navigation in conceptual spaces

Simone Viganò [1✉], Valerio Rubino[1], Marco Buiatti [1] & Manuela Piazza[1]

When humans mentally "navigate" bidimensional uniform conceptual spaces, they recruit the same grid-like and distance codes typically evoked when exploring the physical environment. Here, using fMRI, we show evidence that conceptual navigation also elicits another kind of spatial code: that of absolute direction. This code is mostly localized in the medial parietal cortex, where its strength predicts participants' comparative semantic judgments. It may provide a complementary mechanism for conceptual navigation outside the hippocampal formation.

[1] CIMeC, Center for Mind/Brain Sciences, University of Trento, Rovereto, Italy. ✉email: simone.vigano@unitn.it

Spatial navigation in mammals is based upon a distributed network of regions wherein the medial–temporal lobe is a key node[1]. This region is systematically active when individuals move in real and virtual environments and/or perform spatial tasks[2]. Functional magnetic neuroimaging (fMRI) studies in humans have revealed that this region hosts two types of codes that are crucial for navigation: a distance code (mostly in the hippocampus)[3,4], and a grid-like code (mostly in the entorhinal cortex)[5] that reflect the layout of the environment and of the items and their locations in it. More recently, together with others, we have shown that these types of representations are also recruited in non-spatial contexts, when subjects "navigate" more abstract spaces such that of visual shapes[6], odours[7], people[8] or audiovisual categories[9], even when they are referred to using symbols such as words[10]. The neural machinery that supports spatial navigation in mammals, however, extends well beyond the hippocampal–entorhinal system and includes a larger set of spatially tuned neuronal populations, located in both frontal and parietal cortices[1,2]. For instance, during spatial navigation and orientation it is vital to represent the direction that characterizes movements between any two positions. Previous studies in humans have shown that directional information such as the heading orientation during movement in physical space or the relative direction between two locations in the environment is linked to the activity of the medial parietal cortex[11–16], the parahippocampus[16,17], the thalamus[12,14] and the presubiculum[12,15,18]. The medial parietal cortex, in particular, seems to play a crucial role, because its lesion gives rise to a condition called "heading disorientation"[19] where patients are unable to decide which direction to go to reach a certain goal from landmarks that they otherwise recognize[20]. Interestingly, researchers have identified populations of neurons in the rodent's brain that fire in a heading-selective fashion (so called head-direction cells), both in parietal and subcortical structures such as the thalamus, the striatum, and the subiculum[21,22]. To date, however, no study investigated whether during navigation in a conceptual environment the brain displays signatures of absolute directional information between concepts.

To answer this question, we re-analysed a recent fMRI dataset[10] where participants learned to name 9 novel audiovisual objects, distinguishable for their size and for the sound they produce (Fig. 1a, b), with 9 novel words. To reach this goal they were trained with three different tasks: an association task (where they had to match each object to its correct name), a naming task (in which they typed the name of each presented object), and a semantic comparison task (in which they compared the meaning of the words according to their underlying features, Fig. 1d) (see 'Methods'). Participants were never informed about the geometrical bidimensional arrangement of the stimuli space. Performance at the end of the training was highly above chance in all the three tasks, confirming successful learning (association tasks: accuracy = 92% (std = 9%), chance = ~11%; naming task: accuracy = 91% (std = 10%), chance = ~11%; semantic comparison task: Day 4 = 93% (std = 6%), chance = ~33%).

On the last day of the experiment, subjects participated in a functional MRI session. In the scanner, each trial consisted of a pair of words (e.g., KER and MOS), presented in sequence; sporadically (in ~17% of the trials), they were asked to compare them according to one of their defining features (e.g., "from KER to MOS, *how does the size change? Does it increase, decrease, or remain equal?*" or: "from KER to MOS, *how does the pitch change? Does it increase, decrease, or remain equal?*"). As participants did not know in advance whether the question was going to be asked or not, and, if so, which dimension it was going to relate to, the task implicitly required them to process the full relation between the two words' meanings at each trial. This

relational task could be conceived as eliciting a representation of the direction between word meanings in their underlying feature space (Fig. 1c, d). For instance, the sequential and comparative processing of the words MOS and JOT was conceivable as a movement along a direction at 45° with respect to the horizontal axis (arbitrarily referred as the 0° direction), while presenting the words WEZ and DUN as a movement along a 270° direction. Our aim was to look for brain regions that represented these individual directions in the conceptual space (Fig. 1c, d).

In order to answer this question we used fMRI adaptation[23,24], an approach previously employed to detect the neural representation of faced direction during spatial tasks in virtual environments (e.g., refs. [5,11,14]). Specifically, following Doeller and colleagues[5], we searched for brain regions whose response was suppressed by the repetition of the same direction in the conceptual space. For instance, comparing MOS to JOT elicits the same direction as comparing KER to WEZ. Thus, a voxel containing neurons tuned for specific directions should show a reduction of its activity (adaptation) when a given word pair is preceded by a different word pair that elicits the same direction, and the amount of adaptation should be proportional to how recently in time the word pair sharing the same direction was presented: in direction-coding voxels, the closer in time two identical directions travelled, the stronger the suppression of the fMRI signal (Fig. 2a).

## Results

In order to search for brain regions that represent direction in abstract word space, we used fMRI adaptation by modelling brain activity as a function of the (log) time elapsed since the last presentation of the same direction between words. To account for the fact that the repetition of a given direction between word pairs could sometimes also correspond to the repetition of the very same word pair and/or of the same response type (e.g., "increase in pitch and in size", see 'Methods'), we removed the variance related to two parametric modulators reflecting the (log) time elapsed since the last presentation of, respectively, the same word pair and the same response type.

Then, in order to ensure that the resulting voxels were solely modulated by direction, thus excluding voxels potentially encoding a combination of direction, word pair, and response type, we further masked the map of the adaptation to direction effect with the map of the adaptation to the two other variables. The results revealed a strong fMRI adaptation to direction in a network of regions including the medial parietal cortex, and in particular the precuneus (MNI x,y,z = 0, −54, 62, t = 4.82), the retrosplenial cortex (MNI x,y,z = −6, −50, 0, t = 5.14), the medial superior frontal gyrus (MNI x,y,z = 6, 18, 68, t = 4.89) and the medial portions of the fusiform gyri (right: MNI x,y,z = 30, −80, −14, t = 4.82; left: MNI x,y,z = −30, −76, −18, t = 4.57) extending posteriorly in early visual cortex (MNI x,y,z = 16 −102 −2, t = 4.73) (all p < 0.005, FDR-cluster corrected) (Fig. 2b; Supplementary Fig. 1).

In order to further qualify these results we then performed several control analyses, aiming at answering the following questions:

1. Was the fMRI adaptation to direction solely driven by cardinal directions (0°, 90°, 180°, 270°)? No: we separately modelled cardinal and non-cardinal directions in the first-level GLM analysis and observed a significant adaptation effect at the group-level for both these conditions (all p-values vs zero were <0.05), without significant differences between them (all p-values of the difference "cardinal vs non-cardinal" were >0.36) (Supplementary Fig. 2a; Supplementary Data 1) in all regions.

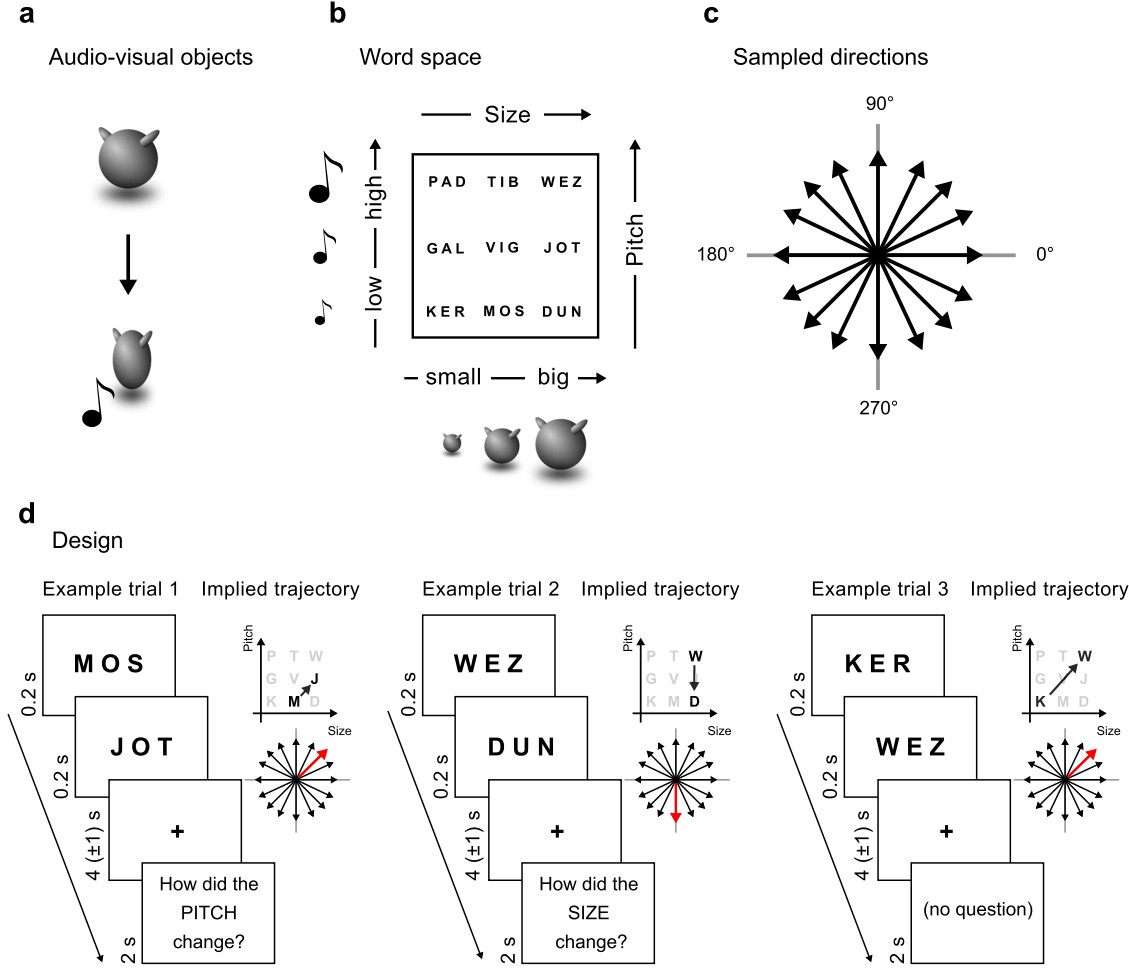

**Fig. 1 Experimental design. a** Example of an audiovisual object, producing a sound during a squeezing animation. **b** Nine audiovisual objects, resulting from the unique combination of size and sound, are labelled with abstract names that participants learn during the course of a behavioural training. **c** In our bidimensional conceptual space, there are 16 possible directions between two words. **d** Examples of trials during the comparison task. For each trial, two words are presented in rapid sequence, and participants are instructed to think about how both the perceptual features implied by the object names changed. This was conceivable as a linear trajectory in the bidimensional conceptual space.

2. Were the direction-responsive brain regions sensitive to changes along each individual dimension? No: we separately modelled, for each trial, the changes along size and pitch in the first-level GLM analysis and observed that none of the direction-sensitive regions was sensitive to changes along size or pitch (parametric regressor modelling, for each word pair, the change along either the size or the pitch axis; the obtained parameter estimates are tested against zero at the group-level; all $p$-values against zero were >0.14) nor they showed a preference for either one of the two sensory dimensions (the obtained parameter estimates for size and pitch are tested against each other; all $p$-values of this comparison "size vs pitch" were >0.20) (Supplementary Fig. 2b; Supplementary Data 1).

3. Could the fMRI adaptation to direction be explained by the degree of correlation with the repetition of word pairs or response type? No: we computed, for each subject and run, the degree of correlation between the regressors modelling the repetition of word pairs or response type (see 'Methods') and the regressor modelling repetition of direction; we then correlated, for each subject, these values to the directional-adaptation signal in the direction-sensitive regions, showing that they were not significantly correlated (all $p$-values > 0.08; in particular, the correlation

between the first regressors pair (word pair x direction) and the directional adaptation in the Precuneus and RSC had $p$-values of 0.3 and 0.08, respectively, while their correlation with the second regressor pair (response type x direction) had $p$-values of 0.46 and 0.89), thus indicating that directional adaptation was not predicted by the degree of correlation between the regressors modelling adaptation to potentially confounding factors and the regressor modelling adaptation to direction (Supplementary Fig. 3; Supplementary Data 1).

4. Could the fMRI adaptation to direction be explained by travelled distance? No: although directions and distances between word pairs did not correlate in our experiment (Pearson's $r = 0.11$, $p = 0.34$), we explicitly tested whether the fMRI signal in the network of regions displaying directional coding was also modulated by distance. Results indicated that none of them also encoded distance, apart from the medial superior frontal gyrus (medial superior frontal gyrus: mean p.e. = 0.47 (std = 0.70), $p = 0.002$; precuneus: mean p.e. = −0.11 (std = 1.57), $p = 0.73$; retrosplenial cortex: mean p.e. = −0.09 (std = 0.59), $p = 0.44$; left fusiform gyrus: mean p.e. = 0.5 (std = 1.28), $p = 0.84$; right fusiform gyrus: mean p.e. = 0.01 (std = 1.29), $p = 0.98$; visual cortex: mean p.e. = 0.14 (std = 1.12), $p = 0.52$; t-tests).

**Fig. 2 Results. a** We modelled the sequence of trials presented during each run taking into account, for each direction implied by a word pair, the elapsed time from the last presentation of the same direction in the conceptual space (see 'Methods'). **b** Brain regions showing adaptation to the direction of movement in the conceptual space during the semantic comparison task, after removing brain regions responding to confounding factors (see 'Methods' and Supplementary Fig. 1A). Results are thresholded at $p < 0.005$, FDR-corrected at cluster level with $q < 0.05$. Group-level effects are plotted onto an average of subjects' structural images. Precu = precuneus; RSC = retrosplenial cortex; Lin = lingual gyrus; mSFG = medial superior frontal gyrus; FG = fusiform gyrus; Visual = visual cortex. **c** Correlation between performance during the semantic comparison task and the adaptation effect in the precuneus ($r = 0.59$) and in the retrosplenial cortex ($r = 0.34$).

While the precuneus and retrosplenial cortex were previously associated with directional coding during spatial tasks[11–16], there were other regions that, in previous studies, were shown to represent heading direction during spatial navigation, such as the entorhinal and parahippocampal cortices, the subiculum, and the thalamus, that were not revealed by our whole-brain approach (e.g., refs. [12,14–18,25]). Moreover, extrastriate regions in Brodmann's area 19 (V5/MT) are known to represent motion direction of visual stimuli[26,27]. Therefore, we used an ROI-based approach (see 'Methods') to search for fMRI adaptation to direction in all these additional regions. The results indicated that among them only the thalamus and BA19 showed absolute direction adaptation (thalamus: mean parameter estimate (p.e.) = −0.19 (std = 0.29), $p = 0.002$; entorhinal cortex: mean p.e. = 0.07 (std = 0.36), $p = 0.15$; parahippocampal cortex: mean p.e. = −0.10 (std = 0.30), $p = 0.10$; subiculum: mean p.e. = 0.045 (std = 0.52), $p = 0.65$; BA19: mean p.e. = 0.18, std = 0.39, $p = 0.02$).

Finally, we reasoned that if the directional signal we observed is truly causally related to mental navigation in conceptual spaces, then we should observe a correlation between the strength of directional adaptation and the accuracy of the semantic comparison task that subjects performed during the fMRI. The results indicated a strong correlation between behaviour and the amount of adaptation displayed by the precuneus (Pearson's $r = 0.596$, $p = 0.001$), and also more moderately by the retrosplenial cortex (Pearson's $r = 0.34$, $p = 0.08$) (Fig. 2c; Supplementary Data 1). The correlation between the precuneus and behaviour remained significant also using non-parametric correlations, more robust to the presence of outliers (Kendall's tau = 0.33, $p = 0.02$), and also after completely removing one potential outlier subject, for which the behavioural accuracy fell more than 1.5 interquartile ranges below the lower quartile (Pearson's $r = 0.48$, $p = 0.01$). None of the other direction-coding regions correlated with behaviour (left fusiform gyrus: Pearson's $r = 0.21$, $p = 0.29$; right fusiform gyrus: Pearson's $r = 0.19$, $p = 0.32$; medial superior frontal gyrus:

Pearson's $r = -0.09$, $p = 0.66$; visual cortex: Pearson's $r = -0.12$, $p = 0.55$).

## Discussion

In this study, we found that a network of brain regions in frontoparietal and ventral occipital cortices represents the absolute direction between word meanings during a comparative task. These regions showed a reduced BOLD response whenever a given direction was repeatedly travelled in abstract space. Among them, the precuneus and the retrosplenial cortex, two areas of the medial parietal cortex typically recruited to represent directional information during spatial navigation and spatial memory[11–16], showed the strongest effects and the highest correlation with the behavioural semantic comparison performance. These results extend previous findings on the recruitment of spatially tuned neural codes for representing conceptual knowledge, mostly limited to the grid-like and distance codes within the hippocampal–entorhinal system, and are in line with empirical evidence showing that the brain navigation system extends well outside the medial–temporal lobe[1,2]. The directional code that we observed here is different from the 6-fold grid-like code previously reported during semantic navigation (e.g., refs.[9,10]). A grid-like code is revealed by an analysis that capitalizes on movement direction to infer the presence of grid-cells in the underlying neuronal population (as first explained in ref.[5]): with this kind of code, directions that are 60° apart from each other are represented similarly (aka cannot be discriminated from each other). The kind of directional code that we show in the present work, however, does not follow any periodic modulation: individual directions are represented separately, as the different directions of a compass. This code is potentially reminiscent of the activity of head-direction cells[21,22], which fire for individual directions. However, (1) the coarse nature of the fMRI signal, (2) the strict link that exists between head-direction (HD) cells and the vestibular system (unlikely recruited in our experiment)[28–30], and (3) the fact that there is only very weak evidence of repetition suppression in the firing of head-direction cells[31] (but see ref.[14] for a discussion on how HD cells adaptation might relate to fMRI repetition suppression) impose extreme caution in driving this interpretation.

A potentially more parsimonious interpretation of our results (which might not be taken in contrast with the previous possibility), is to consider the directional coding here reported as a representation of the relation between the items of a conceptual space: while we and others have shown in the past that this form of relation can be reflected in distance and grid-like codes in the medial–temporal lobe (e.g., refs.[6–10]), now we provide a complementary piece of evidence: it can be conveyed also with a direction code in the parietal and occipital cortices. Reasoning along this more cognitive perspective, a representation of the direction between two concepts during a comparison task is ultimately a representation of how the two concepts differ along their definitional dimensions together; a direction in 2D space thus expresses the ratio between their difference along each dimension: a direction of ~62° (e.g., in our word space equivalent to going from KER to TIB or from GAL to WEZ) implies that the change along one of the two axes (here size, x-axis) is smaller compared to that along the other one (here pitch, y-axis). Holding such a compact representation (of how much one feature changes with respect to another) could be one of the solutions that the brain has developed for compressing multidimensional information, especially when both dimensions are key to solving a comparative task. This could also explain why the strength of this code was predictive of participants' behavioural performance during the semantic comparison task. This correlation was strong in the precuneus, but less so in the retrosplenial cortex, suggesting a possible difference in the functional involvement of these two regions that needs to be further investigated. This code could have evolved from more ancient representational codes dedicated to heading direction during physical movement, and might constitute yet another example of cortical recycling[32].

Contrary to previous studies in the field of spatial navigation and spatial memory, we did not find evidence of this directional code in the medial–temporal lobe. We isolated at least two reasons for this discrepancy. First, the specific methodological approach used in our study, with normalization of fMRI scans to a general template (see 'Methods') as well as the use of a relatively weak magnetic field that could not allow a precise segmentation of hippocampal subfields, might have prevented us to observe directional adaptation in the MTL. Alternatively, our results might indicate that, when processing conceptual spaces, the absolute directional information is mostly represented in brain regions other than the MTL, such as the medial parietal cortex, and that the hippocampal formation is recruited for other representational codes, such as grid-like and distance codes.

To conclude, here we report that a network of regions in the human brain represents the directions existing between elements of a conceptual space. Among them, the precuneus and the retrosplenial cortex seemed to play a crucial role in our experiment. Our results extend the current theoretical accounts that see the processing of the geometry of conceptual spaces as mainly dependent upon the activity of the hippocampal formation and medial-prefrontal cortex, and support the view that the parietal cortex might also play a crucial role in representing relational knowledge between concepts in memory using spatial codes[33,34].

## Methods

**Participants**. Thirty-one right-handed students (21 female) from the University of Trento (Italy), participated in the experiment (mean age: 23.7, std: 3.2). The performance of all but four participants ceiled at over 90% of accuracy at the end of the training; the remaining four participants performed poorly (performance <80% during the final comparison task) and were therefore excluded from the analyses. In total, 27 participants entered the final fMRI analysis, which tops previous similar studies investigating directional coding the field of spatial navigation (e.g., refs.[11–16]) and is in line with a priori estimation of sample size using G*Power (https://stats.idre.ucla.edu/other/gpower, assuming 80% power, alpha = 0.05 and medium effect size of $d = 0.5$, $n = 27$).

**Ethics**. The study was approved by the local Ethics Committee (Comitato Etico per la Sperimentazione con l'essere umano, University of Trento, Italy), in accordance with the Declaration of Helsinki. All participants gave written consent before the experiment. The privacy rights of the participants were observed in accordance with the guidelines of the Ethics Committee of the University of Trento.

**Stimulus space**. Stimuli were 9 novel multisensory objects (Fig. 1a), each named with a distinctive label (Fig. 1b), which were created by orthogonally manipulating the size of a sphere-like shape and the pitch of an associated sound produced during a small squeezing animation. Each of these two dimensions spanned three levels: the objects subtended either 3.75°, 5.73° or 7.64° visual angles and produced sounds at either 500, 750 or 1000 Hz; each object was therefore characterized by one of 9 unique combinations of pitch and size levels. These values were chosen because a previous investigation[9] showed that the resulting audiovisual objects are clearly distinguishable. Stimuli were presented foveally using MATLAB Psychtoolbox (MathWorks) in all experimental phases, at a distance of 130 cm. Object presentation lasted 750 ms. Each word subtended a visual angle of 3.58° horizontally and 2.15° vertically and was presented with black Helvetica font on a grey background. Crucially, the bidimensional arrangement of the stimulus space was never shown to participants.

**Behavioural training sessions**. The experiment consisted of four training sessions and one fMRI session. The neuroimaging session occurred on the day after the last training session. Participants were familiarized with the individual multisensory objects, in random order, before the start of each training session. This familiarization phase consisted in a simple presentation of each object besides its correct name, and participants were asked to simply attend to them. Then, they performed three tasks: an association, a naming and a comparison task. During the fourth and last session, they performed the semantic comparison task only. In none of these

tasks and sessions, participants were exposed to the 2D layout of the semantic space.

**Association task**. During the association task participants were presented with one multisensory object (lasting 750 ms) for each trial, followed by the 9 object names vertically listed in random order. They had to press a number from 1 to 9 on the keyboard to select the name that corresponded to the object. They were immediately shown feedback and correct name in case of wrong answers. The objects were presented in random order, four times each.

**Naming task**. During the naming task, after being presented with one multisensory object (lasting 750 ms), participants were asked to type its name using the computer keyboard. All the three letters were to be typed correctly and participants could not delete typed letters. Once the last letter was typed, participants received feedback as in the previous task. The objects were presented in random order, four times each.

**Semantic comparison task**. In the comparison task (Fig. 1d) participants were presented, for each trial, with two words in rapid sequence, one after the other. Words lasted 200 ms on the screen, with a pause of 250 ms between them. Then, after a reflection period of 4 (±1) s, they were presented with one of these two questions: "how did the size change?" or "how did the pitch change?". Subjects were instructed to respond by pressing one of three buttons according to three options: it increased, it decreased, it did not change. During the reflection period, participants had to mentally consider both features because they could not know in advance which question was about to be presented. The comparison task consisted of 144 trials (all the pairs between different words, repeated twice, once with a question about change in size, once with a question about change in pitch, randomly ordered). On the first training session, participants had no time limit to answer. On the second training session, this was set to 4 s, and to 2 s in the third and fourth ones, to foster automatisation.

**Neuroimaging task**. In the neuroimaging session participants performed the same comparison task as in the last session of training, but the question was only present in a small subsample (16.6%) of trials (because of time limitations in the scanner). This selection was randomized across participants. The experiment was organized in 8 runs of 48 trials each. Participants were explicitly instructed to always think about the meaning of the two words for each trial, because they could not know whether a question would be subsequently presented, nor what dimension it would focus on.

**Trial selection**. There were 72 possible word pairs (pairs of the same word were excluded because they did not subtend any movement), each corresponding to one out of 16 different movement directions (assuming the x-axis as 0°, the possible direction of movements are 0°, 26.5°, 45°, 63.5°, 90°, 116.5°, 135°, 153.5°, 180°, 206.5°, 225°, 243.5°, 270°, 296.5°, 315°, 333.5° (Fig. 1c)). Throughout the experiment, all 16 directions were sampled uniformly with 24 repetitions for each direction, equally divided between runs. Directions and distances of transitions did not correlate ($r = 0.11$, $p = 0.34$).

**Data acquisition and preprocessing**. Data were collected on a 3T PRISMA MRI scanner (Siemens) with standard head coil at the Center for Mind/Brain Sciences, University of Trento, Italy. Functional images were acquired using EPI T2*-weighted scans. Acquisition parameters were as follows: TR = 1 s; TE = 28 ms; FOV = 100 mm; number of slices per volume = 65, acquired in interleaved ascending order; voxel size = 2 mm isotropic. T1-weighted anatomical images were acquired with an MP-RAGE sequence, with $1 \times 1 \times 1$ mm resolution. Functional images were preprocessed using the Statistical Parametric Toolbox (SPM12) in MATLAB following canonical steps: slice timing, realignment of each scan to the first of each run, coregistration of functional and session-specific anatomical images, segmentation, and normalization to the Minnesota National Institute (MNI) space. 7 mm smoothing was applied. Subsequent analyses were performed using SPM12.

**Direction-dependent adaptation**. A word pair presented during a trial implied a trajectory in the underlying conceptual space, with a specific direction. We looked for this information in brain activity using fMRI adaptation (e.g., ref. [5]). We fit a GLM with SPM12 using a parametric modulator that, for each trial, predicted the fMRI signal as a function of the (log) time elapsed since the last presentation of the same direction (see Fig. 2a for a simplified visualization). To guarantee the uniform sampling of all the directions, no trial was excluded from the analysis. This parametric regressor (to which we refer to as "Regressor 3") was complemented and preceded by two additional parametric regressors used for control to potentially confounding factors: Regressor 1 modelled the fMRI activity as a function of the (log) time elapsed since the last presentation of the same word pair. Indeed the same word pair was associated with the same direction. However, each direction was shared across different word pairs, making the two effects dissociable. Indeed, across participants and runs, the correlation between Regressor 1 and Regressor 3

was very small: −0.074; Regressor 2 modelled the fMRI activity as a function of the (log) time elapsed since the same response was planned. Indeed, given our task, the same directions were also associated with the same response, which subjects might have been prepared for even when it was not explicitly required in all trials. However, the same response plans are also shared by trials transversing different directions, indicating that the neural signature for response preparation and directional representations are also in principle dissociable. For instance, trials with a direction of 26° (e.g., KER → JOT) and those with direction of 45° (e.g., VIG → WEZ) require the preparation of the same response (in both cases, both size and pitch increased). Across participants and runs, the correlation between Regressor 2 and Regressor 3 was −0.16. By entering our parametric regressor (Regressor 3) of interest after Regressors 1 and 2 (that is, working on the residuals of these two parametric modulators) we ensured that the signal of the adaptation to the directions was independent to the adaptation to word identities or to response type.

To further isolate brain regions solely sensitive to adaptation to directions we then applied an exclusive masking in SPM, excluding voxels that showed a response to the first two regressors. We then proceeded with a whole-brain analysis. Moreover, in order to verify whether or not the directional signal was present in the other regions of the brain where directional coding had been previously reported (e.g., entorhinal cortex, parahippocampus, subiculum, thalamus), we additionally applied this analysis following an ROI-based approach and selecting ROIs in the thalamus and parahippocampus (masks selected from Pickatlas[35]) as well as in the entorhinal cortex (mask selected from ref. [36]), in the subiculum (corresponding to Brodmann Area 27, selected from Pickatlas[35]), and in Brodmann Area 19 (selected from Pickatlas[35]). Regressor 3 did not correlate with the repetition of the average size or the average pitch implied by word pairs (−0.012 and 0.014, respectively).

**Control analysis on distances**. Although directions and distances between words in our conceptual space did not correlate ($r = 0.11$, $p = 0.34$), we employed an adaptation approach to exclude that the regions displaying sensitivity to directional information were also sensitive to distance information. Following what we and others have previously used to investigate distance codes in conceptual spaces (e.g., refs. [9,10,37,38]), we hypothesized a suppression of the fMRI signal as a function of the proximity of the two words in the underlying feature space (e.g., KER and MOS are closer than KER and WEZ, therefore we expected stronger adaptation in the first case than in the second). Thus, we ran an additional GLM entering the Euclidean distance separating the two words as a parametric modulator. We then extracted the adaptation signal from spheres with a radius of 3 voxels centred on the peaks of the aforementioned direction-sensitive regions: if they also displayed a distance code, then the closer two words are, the more suppressed the fMRI signal in these regions should be.

**Statistics and reproducibility**. Statistically significant effects have been assessed by means of parametric $t$-tests (see 'Results' for the specific parameters).

**Ethics statement**. The study was approved by the local Ethics Committee (Comitato Etico per la Sperimentazione con l'essere umano, University of Trento, Italy), in accordance with the Declaration of Helsinki. All participants gave written consent before the experiment. The privacy rights of the participants were observed in accordance with the guidelines of the Ethics Committee of the University of Trento.

**Reporting summary**. Further information on research design is available in the Nature Research Reporting Summary linked to this article.

## Data availability
In compliance with the guidelines on data sharing and privacy from the Ethics Committee of the University of Trento, the imaging data are available from the corresponding author only for purposes related to the original research question.

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

## Author contributions

S.V., V.R., M.B. and M.P. designed the original experiment; S.V. and V.R. designed and performed the current set of analyses; S.V., V.R., M.B. and M.P. wrote the manuscript.

## Competing interests

The authors declare no competing interests.
