## [Transparent Peer Review File · Communications Biology]

Reviewers' comments:

Reviewer #1 (Remarks to the Author):

Vigano et al report the reanalysis of a previous paper. Prior to scanning subjects learnt the names associated with objects that had different sizes and different sound intensities. During scanning subjects saw object names presented in sequence and were required to consider whether the second object was louder/bigger compared to the first object (although typically no question was actually presented). Using an fMRI adaptation approach they report that the precuneus shows a code that represents the direction between the 2 objects, with activation correlating with subjects performance. They conclude that spatial codes are reused to code multidimensional objects to facilitate task performance.

Overall, I think this is an interesting paper and I have only minor concerns that I think should be addressed.

The authors appear to have reported a directional code in their 2020 Journal of Neuroscience paper, but claim this is a novel finding here. It would be important to be clear as to whether this is the case and the differences between the 2 papers.

Could the overall fMRI adaptation effect be driven by certain cardinal directions (e.g. N,S,E,W)? If these are excluded from the analysis is the effect still present/significant?

Could there be neurons in the precuneus that develop selectivity to the change along each individual dimension? I see that the adaptation effect show as tight a correlation with direction, but can the authors exclude that a contribution from such neurons by explicitly including the change along each individual dimension as 2 separate regressors in the adaptation model?

Why do the authors use an exclusive masking procedure? Wouldn't it be more in keeping with their general approach to include covariates like distance as (earlier) parametric regressors in their model?

It would be worth having a short discussion as to why they don't identify the entorhinal cortex in their analysis given that the Doeller et al., 2010 paper used an adaptation approach based on directionally sensitive entorhinal neurons.

Reviewer #2 (Remarks to the Author):

Viganò et al. provide evidence that the medial parietal cortex shows fMRI adaptation to movement direction in a conceptual space. Human participants underwent fMRI scanning while pseudowords were presented, which the participants had associated with unique audiovisual cues before scanning. These 9 audiovisual cues were defined by the size of visual objects and the pitch of auditory cues, effectively creating a 2D space with the size of the visual object serving as one dimension and the pitch of the auditory cue as the other (3 x 3 design). In each scanning trial, two of the pseudowords were presented in quick succession, which the authors argue corresponds to moving in a certain direction through the conceptual space. They report that various regions showed fMRI adaptation when the movement direction was repeated over successive trials, resembling previously reported adaptation effects shown for virtual navigation (e.g. Doeller et al. 2010) and spatial-orientation tasks (e.g. Shine et al. 2016). The work nicely complements previous reports (including those by the authors: Viganò et al. 2021, 2020) suggesting that the "neural machinery that supports spatial navigation" (line 47) may organize spatial and non-spatial information of any kind in a map-like format. Successive retrieval of closely related features or concepts could thus be considered a form of "navigation in concept space". The manuscript presents a re-analysis of the data of a previous report on this topic (Viganò et al. 2021), it is well written and overall a joy to read. The presented results are clear, and the central

claim of the manuscript is exciting. However, I do have concerns that challenge this claim, mainly surrounding the collinearity between the direction regressor and other experimental factors, and I have some clarification questions, which I hope the authors will be able to address.

The central claim of the manuscript is that the medial parietal cortex shows fMRI adaptation to movement direction in a conceptual space. For this claim to be supported by the data, the direction adaptation must be independent of other factors that may or may not be correlated with direction. Unfortunately, a limitation of the work is that multiple other factors were in fact correlated with direction, which complicates the interpretation of the results.

1) Direction was correlated to some degree with the behavioral response that was given and with the exact pseudowords used for cueing. The authors called such correlations an “unavoidable confound” (line 365) and tried compensating for them by working on the residuals of a nuisance-regression model. However, I still find the interpretation tricky, because the main claim of the paper still builds on collinear regressors, and because orthogonalization does not automatically solve the problem of collinearity. Here are some suggestions that I hope will help to show that the main effect is truly directional, even though the issue remains tricky. A) I recommend visualizing the results obtained for the nuisance regressors (similar to Fig. 2), especially because they were used for exclusive masking of the main effects shown in Fig 2. The latter point should also be explicitly stated in the figure caption and it would be helpful if the mask was shown. B) I believe it would help to quantify the collinearity between regressors. Ideally, the correlation between regressors does not explain the strength of the fMRI adaptation effect across participants (or even within participants across scanning runs). C) Even though the interpretation of the resulting map would be tricky as well, I suggest the authors visualize the main results without exclusive masking and the nuisance regression (one figure) so that the reader can assess the strength and extent of the “raw” adaptation effect, as well as without modeling the trials that drive the collinearity most (another figure). D) The authors show the average fMRI-adaptation effect, but to show that this effect is truly directional it would help if it was visualized for each direction at least for the main regions of interest. Is the adaptation effect driven by all or only by a few directions? Finally, E) One needs to read the details in the methods to know what the authors did when referring to the “hierarchical regression and exclusive masking” (line 112). I recommend adding more details on this to the main text.

2) Unfortunately, direction is also correlated with other experimental factors that were not explicitly modeled, which might be even trickier to control given the study design. For example, not all positions in the 2D space could be approached from all directions, which means that direction correlates with the average size and the average pitch of the audiovisual cues that were recalled last in each trial. Can the authors rule out an influence of this on the main results (maybe again by subsampling the trials somehow)? The adaptation effect would still be interesting, but the problem is that the central claim of the manuscript is that the effect is related to direction. Showing the effect for all directions may help here too.

3) A second more implicit claim of the paper is that the direction adaptation builds on the same neural machinery that supports navigation in physical space. However, what the authors call “navigation” is really a pairwise comparison between two audiovisual cues, or two positions in the space, and it is not clear how participants solve the task. Unlike in physical navigation, it is not necessary in this task to pass over the intermediate positions between two points in space. Therefore, I believe the authors may want to discuss the links to physical navigation more carefully, also because several regions that showed directional signals in physical navigation do not show directional adaptation here (as briefly discussed in the manuscript).

Minor comments:

Which criteria were used to define the response to the nuisance regressors used for exclusive masking?

Head-direction cells do not show adaptation when directions are repeated, speaking against a contribution to the results. I believe this point deserves a short discussion.

Minor typos in line 94 and 403

Reviewer #3 (Remarks to the Author):

Viganò et al. utilized fMRI to investigate semantic directional coding in a bi-dimensional semantic comparison task. They observed repetition suppression for semantic directional coding in some regions previously reported for heading directions in the human brain including retrosplenial cortex, precuneus, and thalamus, but not in previously reported medial temporal regions such as entorhinal cortex or subiculum. Interestingly, they found directional coding in areas that were not previously reported including medial superior frontal gyrus, fusiform gyri, and early visual cortex. The researchers also showed that the directional signal could be modulated by travel distance only in the medial superior frontal gyrus. Further, subjects' task performance was correlated with adaptation signal strength in retrosplenial cortex and precuneus.

This study is well designed, and the findings are clearly reported. Although the data has already been reported in previous publications, the analyses method is very innovative that it answers the new question of semantic directional coding. This study adds value to the emerging field of navigation in a conceptual space and provides more evidence for a general-purposed cognitive map in the human brain. I have several questions especially regarding travel direction and data analyses within medial temporal lobe that need further clarification:

Major comments:

1. More supports and justifications might be needed before naming the semantic directional coding as "absolute travelling direction":

a. It would be clearer if the authors could delineate the differences between previous fMRI studies on heading direction and "absolute travelling direction" to justify why using "absolute travelling direction" is a more appropriate name for the observed directional coding. If the authors aimed to compare semantic spatial coding with the common navigational spatial coding concepts, "heading direction" (combines head direction and travel direction) might be more appropriate than "travel direction" as there were few research evidence specifically studied travel direction in either human or animal neuroscience literatures. Alternatively, "semantic directional coding" might also be more straightforward than "absolute travelling direction".

b. In spatial navigation, travel direction usually relates to body movement that formed a travel trajectory. In the current study, what are the supports for the "movement" component for the conceptual travel? Comparing between two objects has direction indication but is still different from going from one object to another. To find the 'movement' evidence, the authors might consider conducting ROI-based analyses for directional coding in traditionally motion related areas e.g., motor cortex. Regions for traditional direction-selective motion regions such as MT, MST might also worth looking at especially the whole brain analyses already reported directional coding in fusiform gyrus and early visual cortex.

c. "Allocentric" and "egocentric" were typically used to specify frames of reference in spatial navigation. Does the "absolute" in "absolute travel direction" have the same meaning as "allocentric"? If yes, would "allocentric" be a better word than "absolute" here since it's more commonly used? If they are different, it would be better if the difference were elaborated.

d. In line 52 – 55, the fMRI testing phase in both Shine et al. (2016) and Baumann & Mattingley (2010) studies used stationary pictures from environments for detecting heading adaptation. Thus, they may not be a good support for “the representation of heading direction during movement in physical space”.

2. Line 122, early visual cortex has also been reported by previous studies that were associated with directional coding during spatial tasks. Authors might consider citing two papers: Nau et al. (2020) (already cited) and Koch et al. (2020).

Koch, C., Li, S. C., Polk, T. A., & Schuck, N. W. (2020). Effects of aging on encoding of walking direction in the human brain. *Neuropsychologia*, 141, 107379.

3. The authors reported no directional coding in medial temporal regions (e.g., entorhinal cortex, subiculum, parahippocampal cortex). However, additional analyses and explanations might help support the conclusion. The authors normalized brains to MNI space and used atlas for MTL subregions such as subiculum and entorhinal cortex. Would normalizing to MNI space (vs. in native space) drift directional coding signals in MTL subregions? In addition, masks for MTL subregions were traditionally produced through manual segmentation (e.g., Nau et al., 2020). Would manually segmenting MTL subregions be a more rigorous method for analyzing directional coding signals?

4. In the section “Direction-dependent adaptation” starting from line 353, authors mentioned “motor plan”, “motor preparation”, and “motor response”. What does each of the concept refer to? A related question is whether motor cortex should be considered as an ROI?

5. Line 41, authors mentioned that distance code was mostly found in hippocampus. In distance modulation analysis from line 136 to 145, should hippocampal regions be considered as ROIs for testing distance modulation on directional coding?

Minor comments:

1. In figure 2C, figures for precuneus and RSC need to use the same x-axis range.

2. In figure 2C, correlation significance was not included.

3. Authors could consider citing one paper that compares “travel direction” and “head direction” (although only in the entorhinal cortex):

Raudies, F., Brandon, M. P., Chapman, G. W., & Hasselmo, M. E. (2015). Head direction is coded more strongly than movement direction in a population of entorhinal neurons. *Brain research*, 1621, 355-367.

4. Line 385 – 387, the atlas citations Maldjian et al. (2003) and Maass et al. (2015) were not included in the references.

The Reviewers can find our responses to their questions and comments in blue and bold fonts. We would like to thank them for the careful attention they gave to our manuscript and the constructive feedback.

REVIEWERS' COMMENTS:

Reviewer #1 (Remarks to the Author):

Vigano et al report the reanalysis of a previous paper. Prior to scanning, subjects learnt the names associated with objects that had different sizes and different sound intensities. During scanning, subjects saw object names presented in sequence and were required to consider whether the second object was louder/bigger compared to the first object (although typically no question was actually presented). Using an fMRI adaptation approach they report that the precuneus shows a code that represents the direction between the 2 objects, with activation correlating with subjects performance. They conclude that spatial codes are reused to code multidimensional objects to facilitate task performance.

Overall, I think this is an interesting paper and I have only minor concerns that I think should be addressed.

Q1: The authors appear to have reported a directional code in their 2020 Journal of Neuroscience paper, but claim this is a novel finding here. It would be important to be clear as to whether this is the case and the differences between the 2 papers.

A: We would like to thank the Reviewer for considering our manuscript and our previous work. In our J Neurosci study we reported a grid-like “directional” code, namely a very specific modulation of fMRI BOLD signal where movement directions that are 60° apart from each other are represented more similarly than those that are not. This kind of code is considered to be a signature of underlying grid-cells (as explained in Doeller et al. 2010 Nature). What we show in the current manuscript, on the contrary, is the existence of a representation of individual *absolute directions in allocentric space*, as potentially produced by neurons firing selectively for specific absolute directions. We named this code “absolute direction” precisely to disentangle it from the grid-like code observed in our previous reports: let’s consider three exemplar directions: 0°, 60°, and 120° (arbitrarily aligned to the horizontal axis). Because of the specific firing properties of grid-cells (that have a 60° periodicity), a grid-like code would not differentiate among them, as these three directions would be equally aligned or misaligned to the underlying grid firing fields’ orientation. Conversely, an absolute directional code like the one described in the present manuscript, would represent each direction in a different way, thus it would be able to differentiate among them maximally. We acknowledge that the

difference between grid-like and absolute directional code can be made more explicit, therefore we have now added the following sentence to the discussion:

“The directional code that we observed here is different from the 6-fold grid-like code previously reported during semantic navigation (e.g., Viganò & Piazza 2020; Viganò et al. 2021). A grid-like code is revealed by an analysis that capitalizes on movement direction to infer the presence of grid-cells in the underlying neuronal population (as first explained in Doeller et al. 2010): with this kind of code, directions that are 60° apart from each other are represented similarly (aka cannot be discriminated from each other). The kind of directional code that we show in the present work, however, doesn’t follow any periodic modulation: individual directions are represented separately, as the different direction of a compass.”

Q2: Could the overall fMRI adaptation effect be driven by certain cardinal directions (e.g. N,S,E,W)? If these are excluded from the analysis is the effect still present/significant?

A: The overall fMRI adaptation effect doesn’t seem to be driven by cardinal directions, and when they are excluded from the analysis, the effect is still significant. To prove this, in the first-level GLM analysis we separately modelled the trials according to cardinal (Card) and non cardinal (NoCard) directions to see whether the brain regions isolated during our main analyses had a preference for one of the two groups. The results show that in all of the regions previously identified the adaptation to both cardinal and non cardinal directions was significant (all p-values < .05), without significant differences between the two conditions (all p-values > .36): this indicates that our results are unlikely to be driven by some specific directions. This important control is now mentioned in the Results section, and it is added to Supplementary Figure 2A (attached below):

“Was the fMRI adaptation to direction solely driven by cardinal directions (0°, 90°, 180°, 270°)? No: we separately modelled cardinal and non cardinal directions in the first-level GLM analysis and observed a significant adaptation effect at the group-level for both these conditions (all p-values vs zero were < .05), without significant differences between them (all p-values of the difference “cardinal vs non-cardinal” were > .36)(Supplementary Figure 2A) in all regions.”

A

Directional-adaptation effect as a function of cardinality

Supplementary Figure 2 - A. No effect of cardinal direction on directional-adaptation. We verified whether the directional-adaptation signal in the network of regions isolated by our main analysis was mostly driven by cardinal directions (0°, 90°, 180°, 270°). The cardinal (Card) vs non cardinal (NoCard) directions are represented in the top left panel. Parameter estimate of the direction-dependent adaptation effect for the two conditions is plotted for each region. Both cardinal and non cardinal directions elicited significant adaptation (all p-values <.05) in all these brain areas. Cardinal and non cardinal directions did not elicit a different adaptation effect (all p-values >.36).

Q3: Could there be neurons in the precuneus that develop selectivity to the change along each individual dimension? I see that the adaptation effect show as tight a correlation with direction, but can the authors exclude that a contribution from such neurons by explicitly including the change along each individual dimension as 2 separate regressors in the adaptation model?

A: We thank the Reviewer for this interesting question, that was indeed one of the objectives of our previous report on this same dataset (Viganò et al. 2021 NeuroImage). There, we modeled with two separate regressors the change along each individual dimension (size vs pitch) and observed, with fMRI adaptation and a whole brain analysis approach, that changes along size were represented in occipital (visual) cortex (BA18), while changes along pitch were represented in superior temporal (auditory) cortex (Heschl's gyrus and Insula) (please refer to Viganò et al. 2021 NeuroImage - Figures 3D-E). Thus, our previous whole-brain analysis did not reveal any of the direction-sensitive brain regions reported in the present manuscript. However, to exclude any potential overlooked effect, here we repeated the analysis following an ROI-based approach: for each word pair (trial) we modeled the implied change along size and along pitch with two separate regressors at the first-level GLM.

Then, we ran a group-level analysis in the brain regions showing direction-sensitive adaptation. Results demonstrate that:

- 1) No region was sensitive to changes along size nor pitch (parametric regressor modelling, for each word pair, the change along either the size or the pitch axis; the obtained parameter estimates are tested against zero at the group-level; all p-values $> .14$)
- 2) No region showed a preference for either one of the two sensory dimensions (the obtained parameter estimates for size and pitch are tested against each other; all p-values $> .20$).

This indicates that the Precuneus (as well as all other direction-selective regions that we had isolated) does not represent changes along each individual dimension, but rather changes along the two dimensions combined. This information is now summarized in the Results section as well as in Supplementary Figure 2B (attached below):

“Were the direction-responsive brain regions sensitive to changes along each individual dimension? No: we separately modelled, for each trial, the changes along size and pitch in the first-level GLM analysis and observed that none of the direction-sensitive regions was sensitive to changes along size or pitch (parametric regressor modelling, for each word pair, the change along either the size or the pitch axis; the obtained parameter estimates are tested against zero at the group-level; all p-values against zero were $> .14$) nor they showed a preference for either one of the two sensory dimensions (the obtained parameter estimates for size and pitch are tested against each other; all p-values of this comparison “size vs pitch” were $> .20$)(Supplementary Figure 2B).”

B

No effect of changes along size or pitch axes in direction-sensitive regions

Supplementary Figure 2 - B. No effect of changes along size or pitch axes in direction-sensitive regions. We additionally verified that brain regions showing adaptation to direction repetition were not modulated by changes along the individual axes of the feature space (size and pitch). None of the brain regions here isolated showed such an effect (all p-values > .14).

Q4: Why do the authors use an exclusive masking procedure?

A: We applied an exclusive masking procedure to minimize the probability to report brain regions that responded to directional information but also to other features. Parametric regressors and orthogonalization operate on the variance of the fMRI signal but are silent on brain localization: an individual voxel could in principle respond to both the regressors that have been orthogonalized if the variance of its activity captures the variability of both the conditions modelled in the regressors. In our specific design, this could potentially lead to ambiguous results. By additionally applying exclusive masking we wanted to make sure to exclude from the results all those brain regions that show an effect of both the main factor and the confounding factors, and still work on the residuals of the orthogonalization on the remaining parts of the brain. This approach maximizes the probability of reporting brain regions that truly and solely respond to the main regressor of interest (in our case, directional adaptation).

To help alleviate the Reviewers' doubts and for the sake of transparency to the readers, we have also run a whole-brain analysis without the exclusive masking procedure as well as without the parametric modulators (as requested by Reviewer #2). We have added the results in Supplementary

Figure 1B. The results are highly consistent with what we find using our main approach.

B

Adaptation to direction without controlling for confounding factors

Supplementary Figure 1 - B. Adaptation to directions without controlling for confounding factors. We report the results of our main analysis (fMRI adaptation to repetition of the same direction) without exclusive masking (left) and without both exclusive masking and the parametric modulators in the GLM for Regressors 1 and 2 (see Methods). Results are thresholded at $p < .005$, FDR-corrected at cluster level with $q < .05$. Precu = Precuneus; RSC = Retrosplenial cortex; Lin = Lingual gyrus; mSFG = Medial Superior Frontal gyrus; FG = Fusiform gyrus; Visual = Visual cortex.

Q5: Wouldn't it be more in keeping with their general approach to include covariates like distance as (earlier) parametric regressors in their model?

A: We don't think so. The parametric modulators were introduced to eliminate the influence of unwanted confounding factors of no interest for navigation, such as the repetition of words or response type. On the contrary, distance is a relevant variable for (spatial) navigation that potentially co-exists with direction. Therefore, we decided to first remove the variance attributed to non-spatial effects, and then work on the residuals of this regression (and masking) to isolate brain regions representing direction, which was the main objective of this investigation. The potential influence of distance was controlled *a posteriori* not only because it represents a variable that is relevant for spatial navigation (perhaps distance and directions coexist in some brain region, perhaps some brain regions encode distance and direction in a combined fashion), but especially because distance and direction are only modestly correlated by design ($r = 0.11$), as reported in the manuscript. We therefore decided to avoid this approach.

Q6: It would be worth having a short discussion as to why they don't identify the entorhinal cortex in their analysis given that the Doeller et al., 2010 paper used an adaptation approach based on directionally sensitive entorhinal neurons.

A: As explained in the answer to comment 1 (Q1), the entorhinal effect reported by Doeller and colleagues in their 2010 Nature paper refers to the grid-like modulation, where in any given region, the BOLD signal peaks for 6

different direction (recalling grid cells which have not one, but multiple firing fields). Here we report a very different kind of effect: the response is maximal for one single direction.

Although we have used the word “directional” to describe the grid-like code in the past (Viganò et al., 2020), the kind of information conveyed by a 6-fold grid-like code (as the one reported in Doeller et al. 2010 Nature and more recently by us too, albeit using a different method; Viganò et al., 2021) and an absolute directional code (as the one reported in the current work) is qualitatively very different. As we did not attempt to identify the entorhinal grid-cells in this present work, we believe that a discussion about why we did not identify that region is beyond the purpose of the manuscript.

Reviewer #2 (Remarks to the Author):

Viganò et al. provide evidence that the medial parietal cortex shows fMRI adaptation to movement direction in a conceptual space. Human participants underwent fMRI scanning while pseudowords were presented, which the participants had associated with unique audiovisual cues before scanning. These 9 audiovisual cues were defined by the size of visual objects and the pitch of auditory cues, effectively creating a 2D space with the size of the visual object serving as one dimension and the pitch of the auditory cue as the other (3 x 3 design). In each scanning trial, two of the pseudowords were presented in quick succession, which the authors argue corresponds to moving in a certain direction through the conceptual space. They report that various regions showed fMRI adaptation when the movement direction was repeated over successive trials, resembling previously reported adaptation effects shown for virtual navigation (e.g. Doeller et al. 2010) and spatial-orientation tasks (e.g. Shine et al. 2016). The work nicely complements previous reports (including those by the authors: Viganò et al. 2021, 2020) suggesting that the “neural machinery that supports spatial navigation” (line 47) may organize spatial and non-spatial information of any kind in a map-like format. Successive retrieval of closely related features or concepts could thus be considered a form of “navigation in concept space”. The manuscript presents a re-analysis of the data of a previous report on this topic (Viganò et al. 2021), it is well written and overall a joy to read. The presented results are clear, and the central claim of the manuscript is exciting. However, I do have concerns that challenge this claim, mainly surrounding the collinearity between the direction regressor and other experimental factors, and I have some clarification questions, which I hope the authors will be able to address.

The central claim of the manuscript is that the medial parietal cortex shows fMRI adaptation to movement direction in a conceptual space. For this claim to be supported by the data, the direction adaptation must be independent of other factors that may or may not be correlated with direction. Unfortunately, a limitation of the work is that multiple other factors were in fact correlated with direction, which complicates the interpretation of the results.

Q1: Direction was correlated to some degree with the behavioral response that was given and with the exact pseudowords used for cueing. The authors called such correlations an “unavoidable confound” (line 365) and tried compensating for them by working on the residuals of a nuisance-regression model. However, I still find the interpretation tricky, because the main claim of the paper still builds on collinear regressors, and because orthogonalization does not automatically solve the problem of collinearity. Here are some suggestions that I hope will help to show that the main effect is truly directional, even though the issue remains tricky.

A: Before responding to the individual suggestions of the Reviewer, we would like to stress that, aware of this potential problem, we implemented the

multiple regression and exclusive masking approach with the precise scope of eliminating the influence from these confounding factors: by excluding brain regions that show an effect of either word repetition or response type, we in principle make sure that what results is a set of brain regions where activity is likely to be attributable to direction only. In our opinion, the combined use of parametric modulators and exclusive masking made our approach strong enough to support our conclusions. The potential danger of this approach is actually to erroneously report false negatives (namely, overlooking areas truly responding to direction but that fall inside the excluded mask), rather than false positives, a danger that we are aware of and that we accept for the sake of providing reliable results.

We now proceed with a point-by-point reply to his/her suggestions.

Q1-A: I recommend visualizing the results obtained for the nuisance regressors (similar to Fig. 2), especially because they were used for exclusive masking of the main effects shown in Fig 2. The latter point should also be explicitly stated in the figure caption and it would be helpful if the mask was shown.

A: We have now created a **Supplementary Figure 1** to visualize the effects required by the Reviewers. In particular, in panel A we have now reported the results obtained for the nuisance regressors together with the binarized whole-brain mask.

A

Nuisance regressors and exclusive mask

Supplementary Figure 1 - A. Nuisance regressors and exclusive mask. We modeled the sequence of trials presented during each run taking into account, for each direction implied by a word pair, the elapsed time from the last presentation of the same pair (Regressors 1) and the last presentation of a word pair that required the same response preparation (Regressors 2). Results are thresholded at $p < .005$, FDR-corrected at cluster level with $q < .05$. The resulting

activation maps were then binarized and combined to create an exclusive mask for our main analysis and to isolate brain regions that outside this network responded to directional adaptation. I-IPL = left Inferior Parietal Lobule; Prec = Precentral gyrus; MFG = Middle Frontal gyrus; Tha = Thalamus; Crbil = Cerebellum; ITC = Inferior Temporal Cortex; FG = Fusiform gyrus; mSFG = middle Superior Frontal gyrus.

The caption of Figure 2B in the main text is now corrected and states:

“B. fMRI-adaptation results. Brain regions showing adaptation to direction of movement in the conceptual space during the semantic comparison task, after removing brain regions responding to confounding factors (see Methods and Supplementary Figure 1A). Results are thresholded at $p < .005$, FDR-corrected at cluster level with $q < .05$. Group-level effects are plotted onto an average of subjects' structural images. Precu = Precuneus; RSC = Retrosplenial cortex; Lin = Lingual gyrus; mSFG = Medial Superior Frontal gyrus; FG = Fusiform gyrus; Visual = Visual cortex .”

Q1-B: I believe it would help to quantify the collinearity between regressors. Ideally, the correlation between regressors does not explain the strength of the fMRI adaptation effect across participants (or even within participants across scanning runs).

A: We have now computed the degree of correlation between the regressor of interest (modelling repetition of direction) and those of no interest (modelling repetition of word pair and repetition of response type). The results, across subjects and runs, show modest mean correlations: $r = -0.074$ for the former (direction x word pair) and of -0.16 for the latter (direction x response type)

We further verified whether the correlation between regressors could predict the strength of the directional adaptation across participants. We did this control in all the brain regions showing directional adaptation. The results, for the two separate regressor pairs, are as follows:

- for the first regressors pair (direction x word pair), they had a maximum correlation with directional-adaptation (in absolute value for simplicity and across regions) of 0.33 with an associated p-value of .08. This happened in the cluster in the visual area, and all the other regions showed a weaker correlation/higher p-value. In particular, the correlation value in the Precuneus was $r = -0.28$, with a p-value = .14, and the correlation value in the RSC was $r = -0.02$, with a p-value $p = .91$.
- for the second regressor pair (direction x response type), they had a maximum correlation with directional-adaptation (in absolute value for simplicity and across regions) of 0.30 with an associated p-value of .12. This happened in the RSC, and all the other regions showed a weaker

correlation/higher p-value. In particular, the correlation value in the Precuneus was $r = 0.01$, with a p-value of .95.

These results indicate that

- 1) the correlation between the regressors is modest;
- 2) the correlation between the regressors is unlikely to predict the reported directional-adaptation.

We hope that this piece of additional information can help in alleviating the Reviewer's concerns on the interpretability of our results.

To make this information available to the readers, we have added the following sentences to the Methods and the Results, respectively:

"[...] across participants and runs, the correlation between Regressor 1 and Regressor 3 was very small: -0.074 ; [...] the correlation between Regressor 2 and Regressor 3 was -0.16 ."

"Could the fMRI adaptation to direction be explained by the degree of correlation with the repetition of word pairs or response type? No: we computed, for each subject and run, the degree of correlation between the regressors modelling the repetition of word pairs or response type (see Methods) and the regressor modelling repetition of direction; we then correlated, for each subject, these values to the directional-adaptation signal in the direction-sensitive regions, showing that they were not significantly correlated (all p-values $> .08$; in particular, the correlation between the first regressors pair (direction x word pair) and the directional-adaptation in the Precuneus and RSC had p-values of .14 and .91 respectively, while their correlation with the second regressor pair (direction x response type) had p-values of .95 and .12), thus indicating that directional adaptation was not predicted by the degree of correlation between the regressors modelling adaptation to potentially confounding factors and the regressor modelling adaptation to direction."

Q1-C: Even though the interpretation of the resulting map would be tricky as well, I suggest the authors visualize the main results without exclusive masking and the nuisance regression (one figure) so that the reader can assess the strength and extent of the "raw" adaptation effect, as well as without modeling the trials that drive the collinearity most (another figure).

A: We added in Supplementary Figure 1B the results of the adaptation effect without controlling for the confounding factors, which in our opinion are consistent with our main reports and actually confirm that the combination of

parametric modulators and exclusive masking was beneficial to isolate those brain regions likely responding to directions. The figure is attached below.

B

Adaptation to direction without controlling for confounding factors

Supplementary Figure 1 - A. [...] B. Adaptation to directions without controlling for confounding factors. We report the results of our main analysis (fMRI adaptation to repetition of the same direction) without exclusive masking (left) and without both exclusive masking and the parametric modulators in the GLM for Regressors 1 and 2 (see Methods). Results are thresholded at $p < .005$, FDR-corrected at cluster level with $q < .05$. Precu = Precuneus; RSC = Retrosplenial cortex; Lin = Lingual gyrus; mSFG = Medial Superior Frontal gyrus; FG = Fusiform gyrus; Visual = Visual cortex

For what concerns the effect without the modelling of the trials that drive the collinearity the most, we find it hard to identify an objective criterion and threshold for selecting which trials drive the collinearity “the most”. Moreover, as explained in the next response, the special kind of adaptation analysis here implemented (time-dependent) makes selecting specific trials for very fine grained controls difficult: the danger is to have too few observations per condition and to mess up with the temporal order of our time-dependent regressors. Given that we showed in the previous response that the correlation between regressors is modest, we think that this control might not be necessary, and actually that its conclusions might even be misleading.

Q1-D: The authors show the average fMRI-adaptation effect, but to show that this effect is truly directional it would help if it was visualized for each direction at least for the main regions of interest. Is the adaptation effect driven by all or only by a few directions?

A: The Reviewer raises an important point that is also related to our previous response. Unfortunately, our experimental design does not provide enough trials for each direction to perform such analysis: for each individual run of 48 trials we have 16 directions, with each one sampled 3 times; this leads to only 2 time intervals that can be used, within a run, to estimate the degree of adaptation in a voxel for a single individual direction, too few for a statistically reliable analysis.

However, we are able to provide a partial answer to this question by testing whether cardinal directions (0°, 90°, 180° and 270°) might be particularly relevant and drive most of the directional effect, a test also required by Reviewer #1 for this manuscript and proposed as a control in previous studies on the neural correlates of grid-like and directional codes during spatial navigation (e.g. Jacobs et al. 2013 NN; Bellmund et al. 2016 eLife). Grouping directions into 2 families, such as cardinal directions vs non cardinal ones, allows us to test for the putative preference of our direction-sensitive regions for a subset of conditions. The results indicate that this is not the case: in all of the direction-sensitive regions, the adaptation to both cardinal and non cardinal directions was significant (all p-values < .05), without significant differences between the two conditions (all p-values > .36). This important control is now mentioned in the Results section, and it is added to Supplementary Figure 2A (attached below):

“Was the fMRI adaptation to direction solely driven by cardinal directions (0°, 90°, 180°, 270°)? No: we separately modelled cardinal and non cardinal directions in the first-level GLM analysis and observed a significant adaptation effect at the group-level for both these conditions (all p-values vs zero were < .05), without significant differences between them (all p-values of the difference “cardinal vs non-cardinal” were > .36)(Supplementary Figure 2A) in all regions”.

A

No effect of cardinality on directional-adaptation

Supplementary Figure 2 - A. No effect of cardinal direction on directional-adaptation. We verified whether the directional-adaptation signal in the network of regions isolated by our main analysis was mostly driven by cardinal directions (0°, 90°, 180°, 270°). The cardinal (Card) vs non cardinal (NoCard) directions are represented in the top left panel. Parameter estimate of the direction-dependent adaptation effect for the two conditions is plotted for each

region. Both cardinal and non cardinal directions elicited significant adaptation (all p-values <.05) in all these brain areas. Cardinal and non cardinal directions did not elicit a different adaptation effect (all p-values >.36).

Q1-E: One needs to read the details in the methods to know what the authors did when referring to the “hierarchical regression and exclusive masking” (line 112). I recommend adding more details on this to the main text.

A: The following paragraph:

“Using a combination of multiple hierarchical regressions and exclusive masking to account for the fact that the repetition of a given direction could sometimes correspond also to the repetition of the very same word pair and/or of the same response (see Methods), we observed a strong fMRI adaptation to direction in a network of regions including [...]”

has been modified as follows:

“In order to search for brain regions that represent direction in abstract word space we used fMRI adaptation by modeling brain activity as a function of the (log) time elapsed since the last presentation of the same direction between words. To account for the fact that the repetition of a given direction between word pairs could sometimes also correspond to the repetition of the very same word pair and/or of the same response type (e.g., “increase in pitch and in size”. See Methods), we removed the variance related to two parametric modulators reflecting the (log) time elapsed since the last presentation of, respectively, the same word pair and the same response type.

Then, in order to ensure that the resulting voxels were solely modulated by direction, thus excluding voxels potentially encoding a combination of direction, word pair, and response type, we further masked the map of the adaptation to direction effect with the map of the adaptation to the two other variables. The results revealed [...]”.

Q2: Unfortunately, direction is also correlated with other experimental factors that were not explicitly modeled, which might be even trickier to control given the study design. For example, not all positions in the 2D space could be approached from all directions, which means that direction correlates with the average size and the average pitch of the audiovisual cues that were recalled last in each trial. Can the authors rule out an influence of this on the main results (maybe again by subsampling the trials somehow)? The adaptation effect would still be interesting, but the problem is that the central claim of the manuscript is that the effect is related to direction. Showing the effect for all directions may help here too.

A: The Reviewer is correct in noticing that not all positions in the 2D space could be approached from all directions, and that this might influence our results, specifically with respect to the repetition of the average size and average pitch recalled in each trial. To verify whether this could represent an actual confound in our specific design we correlated, for each subject and run, the regressor of main interest (modelling direction repetition) with two additional regressors that modelled repetition of either the average size or the average pitch. The average correlations across subjects were very small: -0.012 and 0.014, for size and pitch respectively. In other words, this indicates that the repetition of a given size (or pitch) value is very unlikely to be correlated with a repetition of a given direction value, thus defending from this alternative explanation of the direction-sensitive adaptation. We have now added this information in the Methods section for the readers:

“Regressor 3 did not correlate with the repetition of the average size or the average pitch implied by word pairs (-0.012 and 0.014 respectively).”

We also believe this is in line with our previous control analysis, also shown in response to Reviewer #1, that the direction-sensitive regions we reported did not reflect changes along size nor pitch (Supplementary Figure 2B).

Q3: A second more implicit claim of the paper is that the direction adaptation builds on the same neural machinery that supports navigation in physical space. However, what the authors call “navigation” is really a pairwise comparison between two audiovisual cues, or two positions in the space, and it is not clear how participants solve the task. Unlike in physical navigation, it is not necessary in this task to pass over the intermediate positions between two points in space. Therefore, I believe the authors may want to discuss the links to physical navigation more carefully, also because several regions that showed directional signals in physical navigation do not show directional adaptation here (as briefly discussed in the manuscript).

A: We agree with the Reviewer that the use of the “conceptual navigation” metaphor might be potentially misleading when directly compared to physical navigation, but we also think that the specific nature of our comparison task makes its use at least reasonable: by asking to compare the meaning of two words to decide about the increment/decrement of some defining features, we believe that it is acceptable to conceive this process as “navigating” in the conceptual space. This mimics the approach used in previous studies in spatial navigation and spatial memory, where static images are used to assess directional coding in the human brain (e.g., Chadwick et al. 2015 Curr Bio). The Reviewer is also right in saying that we don’t know how participants solve the task (although a “*directed* comparison” is intrinsically necessary given the

nature of our question (e.g., from KER to MOS did size increase, decrease, or remain the same?), and that we don't know whether they are mentally passing over the intermediate positions between two points in space. However, we think that even in physical space one doesn't always need to pass through the intermediate points between two locations to *know* their relative direction: while imagining the relative position of our office with respect to our house, we can imagine the correct direction from the former to the latter without mentally reproducing passing through the locations (shops, bars, restaurants, etc.) between them. More in general, and following a more abstract reasoning, given two points in a 2D Euclidean space it is enough to know their x,y coordinates to compute their relative direction (the slope of the line connecting them), without any additional information on the points in between. Finally, the Reviewer is also clearly correct in pointing out that several brain regions typically involved in directional encoding during physical navigation don't show directional adaptation here, and in our opinion this might be related to the fact that physical navigation, differently from "conceptual navigation" requires the integration of our memory and spatial orientation systems with our vestibular and locomotion systems: the case of head-direction cells is paradigmatic of this scenario, as briefly reported in the manuscript.

To sum up, we understand the Reviewer's concerns about our parallelism with spatial navigation, but we think that some of the same doubts could be raised also for fMRI studies involving static stimulation or imagination, that nevertheless succeeded in describing the mechanisms that our brain uses to represent spatial information (distance, direction, etc.). This said, we agree with the Reviewer that more attention could be given when expressing some of these concepts in our manuscript to avoid/resolve ambiguities, especially when their use is mostly metaphorical. Therefore, also following some of the comments of Reviewer #3, we proceeded as follows:

- 1) We removed the reference to "movement" when talking about directions in conceptual space, that could be misleading;
- 2) In the Discussion we made it more explicit that the parallelism with the head-direction system must be taken very carefully, and invite the readers to focus more on the computational/cognitive aspects of the observed directional code;
- 3) We removed the word "travelling" from the title, which could also be misleading;
- 4) We slightly modified the introduction to diminish the focus on the "heading" and "travelling/movement" aspects of navigation. The modified paragraph now states:

“For instance, during spatial navigation and orientation it is vital to represent the direction that characterizes movements between any two positions. Previous studies in humans have shown that directional information such as the heading orientation during movement in physical space or the relative direction between two locations in the environment are linked to the activity of the medial parietal cortex (Baumann & Mattingley 2010; Kim & Maguire 2018; Marchette et al. 2015; Shine et al. 2016; Chadwick et al. 2015; Nau et al. 2020), the parahippocampus (Bellmund et al. 2016; Nau et al. 2020), the thalamus (Kim & Maguire 2018; Shine et al. 2016) and the presubiculum (Kim & Maguire 2018; Vass & Epstein 2013; Chadwick et al. 2015). The medial parietal cortex, in particular, seems to play a crucial role, because its lesion gives rise to a condition called “heading disorientation” (Aguirre and D’Esposito, 1999) where patients are unable to decide which direction to go to reach a certain goal from landmarks that they otherwise recognize (Takahashi et al., 1997). Interestingly, researchers have identified populations of neurons in the rodent’s brain that fire in a heading-selective fashion (so called head-direction cells), both in parietal and subcortical structures such as the thalamus, the striatum, and the subiculum (Taube et al. 1990; Cullen & Taube 2017). To date, however, no study investigated whether during navigation in a conceptual environment the brain displays signatures of absolute directional information between concepts.”

Minor comments:

Q4: Which criteria were used to define the response to the nuisance regressors used for exclusive masking?

A: We used the same threshold of our main analysis: $p < .005$, FDR corrected at cluster level with $q < .05$. This information is now added to the caption of Supplementary Figure 1A.

Q5: Head-direction cells do not show adaptation when directions are repeated, speaking against a contribution to the results. I believe this point deserves a short discussion.

A: We have modified the following sentence in the discussion of our results to include the findings of Taube & Muller 1998 Hippocampus, showing only a very modest evidence of adaptation in the thalamus:

“This code is potentially reminiscent of the activity of head direction cells (Taube et al. 1990; Cullen & Taube 2017), which fire for individual directions. However 1) the coarse nature of the fMRI signal, 2) the strict link that exists between head-direction cells and the vestibular system (unlikely recruited in our experiment) (Blair and Sharp 1996; Brown et al. 2002; Yoder & Taube 2014), and 3) the fact that there is only very weak evidence of repetition suppression in the firing of head-direction cells (Taube & Muller 1998) impose extreme caution in driving this interpretation.

A potentially more parsimonious interpretation of our results (which might not be taken in contrast with the previous possibility), is to consider the directional coding here reported as a representation of the relation between the items of a conceptual space: while we and others have shown in the past that this form of relation can be reflected in distance and grid-like codes in the medial temporal lobe (e.g. Constantinescu et al. 2016; Bao et al. 2019; Viganò & Piazza 2020; Viganò et al. 2021), now we provide a complementary piece of evidence: it can be conveyed also with a direction code in the parietal and occipital cortices.”

Minor typos in line 94 and 403

A: Corrected, thank you.

Reviewer #3 (Remarks to the Author):

Viganò et al. utilized fMRI to investigate semantic directional coding in a bi-dimensional semantic comparison task. They observed repetition suppression for semantic directional coding in some regions previously reported for heading directions in the human brain including retrosplenial cortex, precuneus, and thalamus, but not in previously reported medial temporal regions such as entorhinal cortex or subiculum. Interestingly, they found directional coding in areas that were not previously reported including medial superior frontal gyrus, fusiform gyri, and early visual cortex. The researchers also showed that the directional signal could be modulated by travel distance only in the medial superior frontal gyrus. Further, subjects' task performance was correlated with adaptation signal strength in retrosplenial cortex and precuneus.

This study is well designed, and the findings are clearly reported. Although the data has already been reported in previous publications, the analyses method is very innovative that it answers the new question of semantic directional coding. This study adds value to the emerging field of navigation in a conceptual space and provides more evidence for a general-purposed cognitive map in the human brain. I have several questions especially regarding travel direction and data analyses within medial temporal lobe that need further clarification:

Major comments:

Q1: More supports and justifications might be needed before naming the semantic directional coding as “absolute travelling direction”:

Q1a: It would be clearer if the authors could delineate the differences between previous fMRI studies on heading direction and “absolute travelling direction” to justify why using “absolute travelling direction” is a more appropriate name for the observed directional coding. If the authors aimed to compare semantic spatial coding with the common navigational spatial coding concepts, “heading direction” (combines head direction and travel direction) might be more appropriate than “travel direction” as there were few research evidence specifically studied travel direction in either human or animal neuroscience literatures. Alternatively, “semantic directional coding” might also be more straightforward than “absolute travelling direction”.

A: In our study we reported a representational code that reflects the direction between the items of a conceptual space, so a “*directional* code”. To make explicit the difference between this kind of code and the grid-like one that we and others have reported in the past, and that capitalizes on directions to infer the presence of grid cells from fMRI signal, we named it an “*absolute directional* code”. The adjective “*travelling*” was first introduced to make explicit the kind of comparative experience the task required the participants

to perform (e.g., from KER to MOS, how does the size change?). This adjective has now been removed to make the title less ambiguous, also following the Reviewers' comments. We believe that much of the same ambiguity emerges when we consider the expression "heading" direction: we don't believe the code we reported necessarily represents "heading" direction because that would require an interaction with the egocentric system and/or the vestibular system, for which we don't have any evidence. Such a precise and specific characterization of the directional code here reported is not possible with the current design, therefore to avoid any ambiguity we decide to refer to the code as an "absolute directional code", that better reflects the nature of our result.

Q1b: In spatial navigation, travel direction usually relates to body movement that formed a travel trajectory. In the current study, what are the supports for the "movement" component for the conceptual travel? Comparing between two objects has direction indication but is still different from going from one object to another. To find the 'movement' evidence, the authors might consider conducting ROI-based analyses for directional coding in traditionally motion related areas e.g., motor cortex. Regions for traditional direction-selective motion regions such as MT, MST might also worth looking at especially the whole brain analyses already reported directional coding in fusiform gyrus and early visual cortex.

A: We believe the Reviewer is correct in raising doubts about the too ample use of terms like "movement" or "travel". We agree that a directional coding such as the one we observe here does not necessarily require an active movement to emerge and/or to be accessed, as also explained in the previous response. Throughout the paper we have been using the terms "travelling" and "movement" mainly metaphorically. In this experiment subjects are asked to actively compare the pitch and size of objects referred to by words. This task requires highlighting the relational structure between word meanings in their 2D conceptual space, which can be characterized by a given pattern of distances and directions. Indeed it would be possible to conceive a comparative judgement as a task equivalent to performing a movement but this is by no means necessary. We have therefore rephrased, throughout the manuscript, the strong reference to "movement" and "travelling" direction.

About the use of an ROI in the motor cortex to claim directional information, we are not sure we agree with the Reviewer: while it is true that the motor cortex is involved in locomotion, the direction of movement is independent from the kind of locomotion performed (that is, we use the same locomotor schema during navigation irrespective of our "travelling" direction). For transparency we report to the Reviewer the results of the adaptation analysis in the motor cortex (we used a mask for Brodmann Area 4 taken from Pickatlas), that indeed showed no effect (mean p.e. = -0.03, std = 0.81, p = .83).

Contrary to the motor cortex however, direction-selective regions of the visual cortex are, in our opinion, a more intriguing set of areas to look at. We used as a mask the Brodmann Area 19 (taken from Pickatlas), that should contain V5/MT regions. Here, the effect of our direction-adaptation was indeed significant (mean p.e. = 0.18, std = 0.39, $p = .02$) although significantly weaker than both the effects in the Precuneus ($p = .002$) and in the RSC ($p = .0001$). We now report this finding in the manuscript:

“Moreover, extrastriate regions in Brodmann’s area 19 (V5/MT) are known to represent motion direction of visual stimuli (Maunsell & Van Essen 1983; Kamitani & Tong 2006). Therefore, we used an ROI-based approach (see Methods) to search for fMRI adaptation to direction in all these additional regions. The results indicated that among them only the thalamus and BA19 showed absolute direction adaptation (mean parameter estimate (p.e.) = -0.19 (std = 0.29), $p = 0.002$; entorhinal cortex: mean p.e. = 0.07 (std = 0.36), $p = .15$; parahippocampal cortex: mean p.e. = -0.10 (std = 0.30), $p = .10$; subiculum: mean p.e. = 0.045 (std = 0.52), $p = .65$; BA19: mean p.e. = 0.18, std = 0.39, $p = .02$)”

Q1c: “Allocentric” and “egocentric” were typically used to specify frames of reference in spatial navigation. Does the “absolute” in “absolute travel direction” have the same meaning as “allocentric”? If yes, would “allocentric” be a better word than “absolute” here since it’s more commonly used? If they are different, it would be better if the difference were elaborated.

A: The meaning of absolute in this case is indeed related to that of “allocentric”, as the directional code here reported reflects how items relate to each other independently from the observer’s position. We originally opted for this term because it stressed the fact that directions were represented individually and not, for instance, as a 6-fold grid-like code (as we and others have previously shown in other brain regions).

Q1d: In line 52 – 55, the fMRI testing phase in both Shine et al. (2016) and Baumann & Mattingley (2010) studies used stationary pictures from environments for detecting heading adaptation. Thus, they may not be a good support for “the representation of heading direction during movement in physical space”.

A: We removed this reference from that paragraph.

Q2: Line 122, early visual cortex has also been reported by previous studies that were associated with directional coding during spatial tasks. Authors might consider citing two papers: Nau et al. (2020) (already cited) and Koch et al. (2020).

Koch, C., Li, S. C., Polk, T. A., & Schuck, N. W. (2020). Effects of aging on encoding of walking direction in the human brain. *Neuropsychologia*, 141, 107379.

A: We thank the Reviewer for the suggestion: this reference is now added to the manuscript.

Q3. The authors reported no directional coding in medial temporal regions (e.g., entorhinal cortex, subiculum, parahippocampal cortex). However, additional analyses and explanations might help support the conclusion. The authors normalized brains to MNI space and used atlas for MTL subregions such as subiculum and entorhinal cortex. Would normalizing to MNI space (vs. in native space) drift directional coding signals in MTL subregions? In addition, masks for MTL subregions were traditionally produced through manual segmentation (e.g., Nau et al., 2020). Would manually segmenting MTL subregions be a more rigorous method for analyzing directional coding signals?

A: We agree with the Reviewer that there might be alternative approaches to the one we chose. A stronger magnetic field and a reliable expertise in manually segmenting hippocampal subfields might indeed reveal interesting additional findings. Nevertheless, we don't think that the approach we used here should diminish the reliability of the results. To acknowledge the limits of our conclusion on the null results in the MTL, we have added a paragraph in the Discussion section.

“Contrary to previous studies in the field of spatial navigation and spatial memory, we did not find evidence of this directional code in the medial temporal lobe. We isolated at least two reasons for this discrepancy. First, the specific methodological approach used in our study, with normalization of fMRI scans to a general template (see Methods) as well as the use of a relatively weak magnetic field that could not allow a precise segmentation of hippocampal subfields, might have prevented us to observe directional adaptation in the MTL. Alternatively, our results might indicate that, when processing conceptual spaces, the absolute directional information is mostly represented in other brain regions other than the MTL, such as the medial parietal cortex, and that the hippocampal formation is recruited for other representational codes, such as grid-like and distance codes.”

Q4: In the section “Direction-dependent adaptation” starting from line 353, authors mentioned “motor plan”, “motor preparation”, and “motor response”. What does each of the concept refer to? A related question is whether motor cortex should be considered as an ROI?

A: We apologise with the Reviewer for the confusion: these terms refer to the same kind of phenomenon, namely the fact that some directions required the same kind of response (e.g., “increased”, “decreased”, ...) and therefore could have been represented more similarly because of this. We have now corrected the section to avoid any ambiguity. For what regards the need to conduct and ROI-analysis in the motor cortex, we don't think the manuscript would benefit of such a precise analysis for a confounding variable: the whole-brain results for the response type are now visible in Supplementary Figure 1A, and the results of the direction-adaptation analysis have been provided to the Reviewer when answering Q1b, with an explanation of why we don't think that control should be of particular interest.

Q5: Line 41, authors mentioned that distance code was mostly found in hippocampus. In distance modulation analysis from line 136 to 145, should hippocampal regions be considered as ROIs for testing distance modulation on directional coding?

A: We investigated more thoroughly the representation of distance in our previous manuscript, also focusing on the hippocampus (Viganò et al. 2021 NeuroImage), therefore we believe that the present work should focus on directional coding and its presence or absence in the human brain. An ROI analysis for testing distance modulation on directional coding is surely an interesting suggestion but we believe it goes beyond the scope of the current work.

Minor comments:

Q6: In figure 2C, figures for precuneus and RSC need to use the same x-axis range.

A: The figure has been corrected.

Q7: In figure 2C, correlation significance was not included.

A: The figure has been corrected.

Q8: Authors could consider citing one paper that compares “travel direction” and “head direction” (although only in the entorhinal cortex):

Raudies, F., Brandon, M. P., Chapman, G. W., & Hasselmo, M. E. (2015). Head direction is coded more strongly than movement direction in a population of entorhinal neurons. *Brain research*, 1621, 355-367.

A: We added this reference to the manuscript, thank you.

Q9: Line 385 – 387, the atlas citations Maldjian et al. (2003) and Maass et al. (2015) were not included in the references.

A: **Corrected, thank you.**

Reviewers' comments:

Reviewer #1 (Remarks to the Author):

I thank the authors for thoroughly addressing the concerns i had with the original paper

Reviewer #2 (Remarks to the Author):

Viganò et al. addressed my comments well. They improved the explanation of the analysis and the discussion, visualized the effects observed for the confound regressors and the t-map mask, and they compared the adaption effect across directions, even though only for cardinal vs. non-cardinal directions. Finally, they show that adaptation to average size and pitch of the audiovisual cues does not explain the main effect. All of this is great and helps the reader to interpret the main results. I commend the authors on their efforts and think the manuscript has improved substantially.

The only point I do want to follow up on concerns the inherent correlation among regressors. One of my main concerns was that this correlation could explain the directional adaptation effects that were observed. The authors now tested this by correlating the between-regressor correlation with the strength of the observed directional adaptation across participants, which turned out to be $r = 0.33$, $p = 0.08$ for some ROIs. The paper reports the summarized p-values of these comparisons, concluding there was no confounding effect (because $p > 0.08$). I find this result noteworthy, and to fully convince the reader that the main effects are independent of the raised issue, the results of these tests should be reported more transparently. I recommend reporting the test statistic and scatter plots of the comparisons (regressor correlation vs. adaptation effect for the ROIs) at least in the supplementary material. Again, this is important because the central claim of the paper is that the direction effect is independent of these confound factors.

Reviewer #3 (Remarks to the Author):

I think the authors answered all my questions thoroughly and I have no more concerns.

I also read the authors answers to other reviewers and have some thoughts to share in support of the adaptation of head direction cells. I agree that there is no neuroscience literature specifically reported whether there is adaptation of head direction cells in rodents' brain. It's also technically difficult to measure single cell activities overtime in rodents' brain because neurons are moving. However, research in macaques' brain have observed decreased neural activities in view cells 200 – 400 ms after being exposed to a specific view direction (not published work). Also, in the Shine et al. (2016) paper, the researchers reported that the attenuated BOLD signals in RSC and thalamus towards repeating head direction plateaued over multiple repetitions of the same head direction – this plateau after adaptation makes the fMRI data more consistent with animal electrophysiology.

The Reviewers can find our responses to their questions and comments in blue and bold fonts. We would like to thank them for the careful attention they gave to our manuscript and the constructive feedback.

REVIEWERS' COMMENTS:

Reviewer #1 (Remarks to the Author):

R: I thank the authors for thoroughly addressing the concerns i had with the original paper

A: We thank the reviewer for his/her comments and feedback.

Reviewer #2 (Remarks to the Author):

R: Viganò et al. addressed my comments well. They improved the explanation of the analysis and the discussion, visualized the effects observed for the confound regressors and the t-map mask, and they compared the adaptation effect across directions, even though only for cardinal vs. non-cardinal directions. Finally, they show that adaptation to average size and pitch of the audiovisual cues does not explain the main effect. All of this is great and helps the reader to interpret the main results. I commend the authors on their efforts and think the manuscript has improved substantially.

The only point I do want to follow up on concerns the inherent correlation among regressors. One of my main concerns was that this correlation could explain the directional adaptation effects that were observed. The authors now tested this by correlating the between-regressor correlation with the strength of the observed directional adaptation across participants, which turned out to be $r = 0.33$, $p = 0.08$ for some ROIs. The paper reports the summarized p-values of these comparisons, concluding there was no confounding effect (because $p > 0.08$). I find this result noteworthy, and to fully convince the reader that the main effects are independent of the raised issue, the results of these tests should be reported more transparently. I recommend reporting the test statistic and scatter plots of the comparisons (regressor correlation vs. adaptation effect for the ROIs) at least in the supplementary material. Again, this is important because the central claim of the paper is that the direction effect is independent of these confound factors.

A: We thank the reviewer for his/her comments and feedback, and we also agree on the recommendation of adding a Supplementary Figure to better visualize the mentioned results. This can be now found as Supplementary Figure 3, that we also attach here. Please notice that in reviewing these

correlations we noticed some minor errors in the scores reported, that are now corrected in the main text.

A

The correlation between word-pairs and direction repetition does not predict directional adaptation

B

The correlation between response type and direction repetition does not predict directional adaptation

Supplementary Figure 3 - A. The correlation between word-pairs and direction repetition does not predict directional adaptation. We verified whether the directional-adaptation signal in the network of regions isolated by our main analysis was correlated with the degree of collinearity between the regressor modelling repetition of word-pair and the one modelling repetition of direction. **B. The correlation between response type and direction repetition does not predict directional adaptation.** We verified whether the directional-adaptation signal in the network of regions isolated by our main analysis was correlated with the degree of collinearity between the regressor modelling the response type and the one modelling repetition of direction.

Reviewer #3 (Remarks to the Author):

R: I think the authors answered all my questions thoroughly and I have no more concerns.

I also read the authors answers to other reviewers and have some thoughts to share in support of the adaptation of head direction cells. I agree that there is no neuroscience literature specifically reported whether there is adaptation of head direction cells in rodents' brain. It's also technically difficult to measure single cell activities overtime in rodents' brain because neurons are moving. However, research in macaques' brain have observed decreased neural activities in view cells 200 – 400 ms after being exposed to a specific view direction (not published work). Also, in the Shine et al. (2016) paper, the researchers reported that the attenuated BOLD signals in RSC and thalamus towards repeating head direction plateaued over multiple repetitions of the same head direction – this plateau after adaptation makes the fMRI data more consistent with animal electrophysiology.

A: We thank the reviewer for his/her comments and feedback, and in particular for pointing out the interesting results and discussion of the Shine et al. 2017 JNeurosci paper. We now highlight that work in the discussion of our Manuscript when discussing the putative link with HD cells.